# The Synergistic Effects of Hybrid Micro and Nano Silica in Influencing the Mechanical Properties of Epoxy Composites—A New Model

**DOI:** 10.3390/polym14193969

**Published:** 2022-09-22

**Authors:** Raja Nor Raja Othman, Dinesh Kumar Subramaniam, Nursa’adah Ezani, Mohamad Faizal Abdullah, Ku Zarina Ku Ahmad

**Affiliations:** Department of Mechanical Engineering, National Defence University of Malaysia, Kem Sungai Besi, Kuala Lumpur 57000, Malaysia

**Keywords:** hybrid silica particles, nano silica, micro silica, synergistic effect, extended model for hybrid

## Abstract

**Highlights:**

**Abstract:**

Epoxy nano composites containing micro and nano silica were prepared by varying the filler’s weight loading as an attempt to investigate the effects of incorporating these fillers in influencing its mechanical properties. Mechanical properties characterizations include the evaluation of tensile. The mechanical properties of the epoxy composites were found to tremendously increase as both micro and nano silica were added together at a 1:1 wt.% ratio. For example, the highest values of Young’s modulus were recorded to be 5.39 GPa for 25 wt.% loading (12.5 wt.% Micro + 12.5 wt.% nano), while Young’s modulus values of 5.22 MPa and 5.32 MPa were recorded for micro and nano silica, respectively, at the same weight loading. The most outstanding results were observed at 25 wt.% hybrids (12.5 wt.% micro silica + 12.5 wt.% nano silica), where the values of Young’s modulus were increased by 228% compared to the neat epoxy. This study successfully demonstrated synergistic effects demonstrated by combining micro and nano silica fillers, which created an interaction that significantly enhanced the Young’s modulus of epoxy composites.

## 1. Introduction

Epoxy is used widely in a broad area of technology. It can be formulated to provide strong bonding in various types of substrates over a vast range of conditions. It can be tailored to desired properties of materials [1]. Other than that, epoxies had very useful properties, such as high modulus, high service temperature, and no creep due to their highly crosslinked structure, which are widely used as adhesives, coatings, and matrices of reinforced composites. However, epoxies are inherently brittle because they had very poor resistance to the initiation and propagation of cracks from the defects [2]. Tengke Ye et al. mentioned that the size of fillers is expected to play a crucial role in the type of toughening mechanisms filled in epoxy-based composites and the small size particles are superior to those that are larger due to the compact interface and larger tension of the movement dislocations [3].

The presence of these nano particles as fillers increases Young’s modulus and yield strength better than microparticles. This is because nano particles are relatively small and could fill in the space within the chain link, which can cause the toughness, cyclic-fatigue resistance, and viscosity of the polymer to increase. Peerapan Dittanet studied nano silica with different ranges of filler content, which are 23, 74, and 170 nm. The result shows that the value of Young’s modulus K_IC_ and G_IC_ increased from the value of neat epoxy. The 170 nm size nano silica particles showed the highest Young’s modulus among other size of nano particles. The improvement was 5.78 GPa, which is 40% higher than neat epoxy. The other particle sizes are observed to follow the same trend. However, it is concluded by the author that the size of the particle did not exhibit any affect on the value of Young’s modulus [4]. Bagherzadeh et al. studied the mechanical and bonding properties of micro/nano fillers containing epoxy adhesives for anchoring steel bars in concrete. They concluded that using micro silica particles, with a mesh size of 600, the composites showed 266%, 205%, 60%, and 102% improvement in the compressive strength, compressive modulus, tensile and flexural, respectively. Additionally, they also stated that large micro silica particles to epoxy binder led to lower fillers to polymer interfacial interaction [5]. It is seen from the previous study there are several factors, such as the size of particles and the volume percent of fillers affect the fracture toughness of the epoxy composites [6]. In a separate study, Tadaharu Adachi et al. showed the effect of particle size and volume fraction on strength of spherical silica particle-filled epoxy composite. They performed an experiment using silica nano particles with a diameter ranging from 1.56 µm to 240 nm, and volume fraction ranging from 0 to 0.35 vol.%. They reported a tremendous increment in G_IC_ values from 253 J/m^2^ for the epoxy to 1124 J/m^2^ with 240 nm silica size. In conclusion, the researcher mentioned that when the size of the filler has increased the value of G_IC_ and Young’s modulus also increase constantly [7].

The approach of incorporating more than one filler into polymers has been used by several researchers to improve the mechanical properties of the composites. This is due to the synergistic effects in composites, introduced by more than two fillers. For example, Gokuldass et al., studied the effect of hybrid composite tailored with glass, Kevlar fibre-reinforced in nano silica and micro rubber blended epoxy. The epoxy was modified with the addition of 9% micro rubber and 11% of nano silica by weight fraction. The result showed tensile strength at a content of 11 wt.% of nano silica increased tensile and flexural modulus. The highest tensile and flexural strengths obtain are 275 and 162 MPa, respectively. The tensile strength showed and improvement of 40% from the neat epoxy, whereas for the flexural strength with the addition of nano silica showed an improvement of 87%. The researcher reported that nano silica gives a better interfacial bonding than other composites [8].

S. Mutalikdesai et al. examined glass reinforced epoxy hybrid composites using fly ash, nano clay and zinc oxide as fillers. They used the hand layup technique and tested for mechanical characterization for these hybrid composites. Single fillers with 3 wt.% of nano clay showed 400 Mpa, whereas when single fillers with the same amount of fillers of fly ash showed 360 Mpa, but when hybrid fillers are used the ultimate tensile strength increases to 584 Mpa, which increases approximately 38% and 32%, respectively from the single fillers. They conclude this is due to the great influence of filler dispersion [9].

The effect of nano silica particles with reinforced carbon, Kevlar, and epoxy resin (CKFRE) was studied by Alsaadi et.al. It is recorded that the tensile strength was increased from 371.74 MPa (single filler) to 444.98 MPa (dual fillers), resulting in 46.8% improvement while modulus showed 31% improvement compared with unmodified CKFRE samples. This shows that dual fillers have an extra advantage over single fillers where the addition of nano silica had an improved brittle nature, maximum elongation, and their strength at breaking point. They also concluded that nano silica incorporation with epoxy enhance the interfacial bonding strength and increase the load transfer between composites [10].

Studies reported in the literature suggested that the size of the silica nano particles played an important role in enhancing the mechanical properties of the composites. An interesting observation was also reported as a result of incorporating two types of fillers, of different sizes. As such, this study tends to investigate the synergistic effects of adding two types of fillers, which are nano and micro scale, into an epoxy polymer. The study will be more focused on Young’s modulus which will be measured for composites containing both single and dual fillers with different weight percentages. The obtained result will be compared with an available model, such as Mori-Tanaka, Halpin Tzai, and ROM. The results presented are expected to help us further understand the synergistic effects demonstrated by these dual filters of different sizes. Furthermore, from the obtained result of hybrid and comparing to the available model of single filler this paper is moving one step closer to full fill the gap of producing a model for the hybrid composition of silica.

The model of Mori-Tanaka is adopted in this model with the addition of a 3rd phase, which is the second filler. The model is adjusted so that it helps to predict epoxy silica composite with more than one filler. The model used comprise of Poisson ratio, volume fraction of fillers, and Young’s modulus of the fillers and matrix, which are epoxy. The model could predict up to a 90% approximate value of the hybrid fillers.

## 2. Experimental Details

### 2.1. Materials

The epoxy resin used in this study is DGEBA epoxy (Oriental Option, Penang, Malaysia) with the code CP812P Part A (Epoxy) (Oriental Option, Penang, Malaysia), which has a viscosity of approximately 8.0 ± 2.0 and mixing ratio of 100. For the CP812P (Part B) hardener, it has viscosity of approximately 0.03 ± 0.02 and the mixing ratio is approximately 50. Both epoxy and hardener are produced by a local manufacturer. The cure time for this type of epoxy is at least 17 h at room temperature. The ratio of epoxy to hardener was 2:1 for each of the samples as recommended by the manufacturer. The specification of silica for both sizes is as shown in Table 1:

**Table 1 polymers-14-03969-t001:** Properties of Micro and Nano Silica [11,12].

Specification	Silica	
	Micro	Nano
Grade	High Purity	Nano
Mfr	Sigma Aldrich, Germany	Elite–Indus, China
Particle Size	40–75 µm	30 ± 5 nm
Pore Size	0.7–0.9 cm^3^/g	-
PH	7	5–7
Purity	99.8–99%	99.8–99.9%
Surface Area	450–550 m^2^/g	220 ± 30 m^2^/g
Young’s modulus	70 GPa	
Poisson ratio	0.17	

### 2.2. Fabrication and Characterization

The epoxy resin is formulated by diluting the epoxy with the silica to produce a sample in the range of 0–35 wt.%. The mixture is mixed well at 500 rpm for 5 min using a mechanical stirrer, IKA-WERKE GMBH & Co. KG, Germany. Then hardener is added with the ratio of 2 to 1 of epoxy and hardener. Then the mixture is poured into a mould for curing at room temperature for 24 h.

Tensile tests are carried out using Universal Testing Machine INSTRON 5569 A, Instron, United Kingdom, Buckinghamshire. Tensile test was carried out according to ASTM D638 standard with load applied constantly at the rate of 2 mm/min. The specimen for the tensile test was produced in a dog bone shape with the dimension of 13 mm width, 170 mm length, and 3 mm thickness. For a dual filler, similar sample preparation and characterization are performed. Both fillers were first added to the epoxy at a 1:1 wt.% ratio. The experiments are conducted at total fillers loadings of 0–35 wt.%.

## 3. Result and Discussion

### 3.1. Morphology Analysis

The surface morphology of nano silica and micro silica composites were characterized using Field Emission Scanning Microscopy (FESEM) Zeiss Gemini 500, Carl Zeiss Microscopy, Germany. As shown in Figure 1, the presence of micro and nano silica particles is seen at the surface. Figure 1 below shows the difference in morphology between micro and nano silica. It is obvious that size of the micro silica particle is between 40 µm to 75 µm, as specified by the manufacturer. However, the size of the nano particles cannot be directly determined as they appear to agglomerate due to Van der Waals attraction.

### 3.2. Micro Fractography of the Composite

FESEM observation in Electron Dispersive X-ray (EDX) mode is also conducted on the fractured sample of the composites to check the distribution of the silica fillers. Figure 2 below shows the morphology of the fractured surface of 5 wt.% hybrid (A–C) and 30 wt.% hybrid (D–E). It can be clearly seen from the figure that the presence of micro silica in spherical shape for both loadings, with more particles are seen for the fractured 30 wt.% samples. The elemental analysis further confirms that the spherical particles are indeed silicon and oxygen for 5 wt.% hybrid (B–C) and 30 wt.% hybrid (E–F). Besides that, the presence of silicon and oxygen can also be seen throughout the areas, which could be coming from the presence of nano silica. The images reveal that both micro and nano silica are homogenously distributed throughout the composites. The weight percent analysis is also shown in Figure 2. Obviously, higher distribution for silicon and oxygen is recorded for samples containing 30 wt.% fillers, compared to samples containing 5 wt.% fillers.

### 3.3. Results of Mechanical Properties

The effect of silica particles in epoxy also can be seen in other mechanical properties obtained from the tensile test, such as ultimate tensile strength, proportional limit, yield strength and rupture strength, and these properties show the ability of the fillers towards epoxy composite. Hybrid exhibited better properties then the single fillers. The increase in properties is as expected as the modulus goes higher for the composite. Figure 2 shows the different between micro, nano and hybrid silica properties.

In addition, the homogenous dispersion of the high stiffness among the fillers in the system or matrix enhanced the toughness of the system which can be calculated by the larger area under the curve. When the tensile load increases, the material tries to elongate in its normal way. However as seen in the curve silica fillers resist deformation as the bonding between particles are strong and supporting one and another. The mechanical properties of the hybrid silica show a good improvement than the single filler, which enhances the epoxy composite to a stronger bonding. This was also mentioned by Aidah et al. where the effect of adding a nano silica filler in an epoxy composite produced promising mechanical properties that enhance the strength and modulus by approximately 38% and 24%, respectively, for 25 wt.% of loading compared to neat polymer without scarifying failure strain [13]. Muhannad et al., studied on the properties of the nano composite, which produced similar promising result where the Young’s modulus and flexural strength increase with increasing volume fraction of fumed silica nano particles, which concluded due to complicating crosslink between polymer chains [14]. Crosslinking is also studied by C. Sperandio et al. the effect of crosslink rate on silica with epoxy resin made significant changes during heat transfer and curing process. It was concluded that thermal and mechanical properties of epoxy resin are highly dependent on the crosslinked three dimensional microstructure formed during the curing process [15].

### 3.4. Young’s Modulus Result

Table 2 shows the values of Young’s modulus for silica fillers filled with epoxy as a function of loading (wt.%) for micro silica, nano silica, and their hybrids. For single filler, it could be seen that both nano and micro silica show lower values of Young’s modulus at the early stage but continue to increase with respect to increment in silica loading than the neat epoxy.

For example, the value of Young’s modulus for composite containing micro silica decreased to its lowest value, which is 1.35 GPa at 0.05 wt.% loading, compared to 2.36 GPa recorded for neat epoxy. Similarly, the value of Young’s modulus for composite containing nano silica also reduced tremendously to 0.97 GPa for 3 wt.% loading, which is more than a 100% reduction compared to the neat epoxy, however it increases with the increment of silica loading. 

A similar trend is also reported by Koh et al. where Young’s modulus decreases when the loading is increased [16]. This is expected due to the lack of interface bonding between the silica particles and with epoxy matrix. It is seen that dispersion and interaction are the main factors that may influence the mechanical properties of composite materials. The presence of bubbles between the particles may also affect the strong bonding between the particle, which will result in inaccurate measurement of the mechanical properties of the materials. 

However, the hybrid composite of micro and nano silica fillers showed rather interesting findings. At 1 wt.%, dual fillers composites (0.5 wt.% nano + 0.5 wt.% micro) recorded a Young’s modulus value of 2.21 GPa, compared to 1.43 GPa for 1 wt.% micro silica, and 0.99 GPa for 1 wt.% for nano silica. Another remarkable result is obtained from composites containing 25 wt.% loading (12.5 wt.% Micro + 12.5 wt.% nano), where the Young’s Modulus value was measured to be 5.39 GPa, compared to 4.68 GPa for 25 wt.% micro silica and 5.00 GPa for 25 wt.% nano silica. The highest value of Young’s Modulus was 5.39 GPa, which was measured at 25 wt.% of hybrid-silica, which is a 228% improvement, compared to neat epoxy. However, the hybrid silica reaches the highest value of Young’s modulus with less silica loading compared to a single filler, which shows the bonding incorporating different sizes of silica creates a stronger bond than single filler. These results clearly showed an excellent synergistic effect, demonstrated by incorporating nano silica and micro silica in enhancing the values of Young’s modulus of the epoxy matrix. Azmi et al. investigated rubbery and nano hollow glass sphere particles in composites and observed that Young’s modulus improved as the dual filler loading increased [17]. It is seen that the particle is strongly bonded which prevents their mobility and deformation of the matrix by introducing mechanical resistance. This is also due to the effect of particle size of nano silica filled in epoxy polymer where smaller nano particles could fill in the blanks and increase the interfacial area between the silica particle surface and polymer matrix [18,19].

Figure 3 shows the effect of silica when introduced in epoxy, which shows an increment in a constant manner with the increment of silica loading. The modulus clearly shows as expected where the addition of silica particles with a higher modulus will increase the strength of the sample compared to neat epoxy, which has a lower modulus. Note that for all silica particle composite micro, nano, and hybrid the modulus increases with silica loading. As reported, earlier size of the single filler in the epoxy matrix does not affect the properties of Young’s modulus. However, a combination of micro and nano silica fillers affects the properties as the modulus increases rapidly with the loading. Therefore, the current study is in agreement with the literature review where combination particles in an epoxy matrix affect Young’s modulus. The increase of modulus is influenced by the interaction between two phases of particles, which tighten the matrix by filling in the blank or filling the empty spot when the different size of a particle is combined in the epoxy matrix creating a stronger bonding [20]. This stronger bonding restricts the mobility of particles and the deformation of the whole matrix by introducing mechanical restraint. Figure 4 shows the comparison between the micro, nano and hybrid fillers.

### 3.5. Comparison with Available Model

Many proposed models could predict theoretically the properties of the polymer that incorporated particulate filler. Einstein model was an early model that was used to predict the shear viscosity of dilute suspension containing rigid spheres. The model then is enhanced to predict Young’s modulus. In this paper mainly four models are evaluated to predict and compare the values produced by the experiment with standard deviation calculated with at least three samples each. The Mori-Tanaka, Halpin-Tsai, Kerner, and Rule of Mixture (ROM) models are considered popular in polymers and most relevant for the particulate filler polymers [13].

The Mori-Tanaka model is widely used by researchers in polymer fill to predict Young’s modulus as a function of volume fraction and particle geometry. In this model, it is assumed only two phases exist which are matrix and reinforcement of fillers and are perfectly attached. Mori- Tanaka approach is advised to be useful to predict the overall properties of composites that have a scale of micrometre or larger. The parameters used for the fitting are in Table 3.

**Table 3 polymers-14-03969-t003:** Material parameters used in prediction and modelling.

Parameters	Symbol	Value	References
Bulk modulus of epoxy matrix, GPa	K_m_	5.23	Equation (2)
Bulk modulus of silica microparticle, GPa	K_f1_	35.35	Equation (3)
Bulk modulus of silica nano particle, GPa	K_f2_	35.35	Equation (3)
Shear modulus of epoxy matrix, GPa	G_m_	1.24	Equation (6)
Shear modulus of silica microparticle, GPa	G_f1_	29.91	Equation (5)
Shear modulus of silica nano particle, GPa	G_f2_	29.91	Equation (5)
Young’s modulus of epoxy matrix, GPa	E_m_	3.45	Table 1
Young’s modulus of micro silica particle, GPa	E_f1_	70	[4]
Young’s modulus of nano silica particle, GPa	E_f2_	70	[21]
Poisson ratio of epoxy matrix	V_m_	0.39	[4]
Poisson ratio of silica nano particle	V_f1_	0.17	[4]
Poisson ratio of silica microparticle	V_f2_	0.17	[4]

Halpin-Tsai model is also widely introduced by researchers in predicting polymer properties. This model considers modulus of filler, E_f_ and epoxy matrix E_m_ as well as incorporated with shape factor. Therefore, it is found to be more accurate with the fillers of nano tubes, nano clay, and nano sphere. Kerner model was generalized by Lewis and Nielsen, which derived from a geometry that depends on the passion ratio of both matrix and a constant that is given to modify the model according to the shape of the fillers.

Lastly is the ROM model which comprises two types of equations for transverse loading and perpendicular loading with a simple understanding that combines the modulus of both fillers and epoxy with respect to the volume of fillers. Table 4 shows the models and supporting equation of the models to predict the value of Young’s modulus.

**Table 4 polymers-14-03969-t004:** Mori-Tanaka, Halpin-Tsai, Kerner and ROM models.

Name of Model & References	Model	Remarks
Einstein [22]	$\frac{E_{c}}{E_{m}}=1+2.5\;V_{f}$	Assumed perfect adhesion between filler and matrix and no interaction between particles. Valid for spherical particles.
Kerner [23]	$\frac{E_{c}}{E_{m}}=\frac{1\;+\;ABV_{f}}{1\;-\;B\psi V_{f}}$	Showed the closest agreement with experimental data for silica epoxy nano composite.
Mori Tanaka [24]	$E_{c}=\frac{9{\bar{K}}_{c}{\bar{G}}_{c}}{3{\bar{K}}_{c}\;+\;{\bar{G}}_{c}}$	Assumed that only two phases exist which are matrix and reinforcement and perfectly bonded together. Can accurately predict overall properties of composites when reinforcement is in the micrometer or larger scale.
Halpin Tsai [25]	$\frac{E_{c}}{E_{m}}=\frac{1\;+\;\varsigma \eta \phi }{1\;-\;\eta \phi }$	Can give an accurate prediction for carbon nano tubes, nano clay and nano sphere silica-filled systems.
ROM	$E_{2}=\frac{E_{m}E_{f}}{V_{m}E_{f}\;+\;V_{f}E_{m}}$	Perpendicular models are used to obtain the result for Young’s modulus; the pull theory is used.

*E_c_* is Young’s modulus of the nano composite, E_m_ is Young’s modulus of the matrix, V_f_ is the volume fraction of the filler, s is the crowding factor of nano particle that is related to interparticle interactions, p is the aspect ratio of particles, A and B are constants and ψ is the maximum packing fraction of filler in the matrix. K = bulk Modulus, G = Shear Modulus, ϕ=sphere volume fraction and ς=2a where a=1 for sphere.

The comparison is made with the different available models along with standard deviation which is calculated from at least 3 samples. Table 5 shows the error R^2^ calculated between the model and the best fit model that could be used to predict the silica composite.

The regression of R^2^ clearly shows that all the models could agree with the value of Young’s modulus for single filers. But for a hybrid filler, the model differs where the prediction does not include the second filler. All the models can be used to describe the value of Young’s modulus obtained from experimental data. The Figure 5 shows the comparison between theoretical and experimental values. From the comparison, it is seen that all the models fit the single fillers for prediction of Young’s modulus; however, for hybrid fillers it differs than single fillers based on the value of regression R^2^ from ROM, Halpin Tsai and Mori-Tanaka models.

### 3.6. Modelling Studies

There are many models proposed to predict single fillers for silica however the hybrid does not. This paper is intended to fill the gap by modifying and optimizing the best model that could fit the experimental data as it could be used for predicting the value of the Young modulus. The basic idea was to add and alter the available model by adding the character and properties of the second filler into the equation. The model used to modify to suit the hybrid model is Mori-Tanaka as this model comprises the two phases which are matrix and reinforcement that is adopted to suit more than one filler composite by adding the character of the second filler as the first. The dual filler Young’s modulus for reinforced composite could be predicted by the relationship
(1)$$E_{c}=\frac{7.2\;K_{c}\;G_{c}}{\left(2.4-K_{c}\right)+G_{c}}\;$$
where *E*, *K* and *G* are Young’s modulus, bulk modulus, and shear modulus, respectively. The subscript *C* stands as a composite, whereas the value of *K_C_* and *G_C_* can be solved as below:(2)$$K_{m}=\frac{\;E_{m}}{1.5\left[5.5-\left(\frac{E_{m}}{G_{m}}\right)\right]}\;$$
(3)$$K_{f}=\frac{\;E_{f}}{1.5\left[5.5-\left(\frac{E_{f}}{G_{f}}\right)\right]}\;$$
(4)$${\bar{K}}_{c}=K_{m}+\left[\frac{V_{f1}K_{m}\left(K_{f1}-K_{m}\right)}{K_{m}+\beta_{2}\left(1-V_{f1}\right)\left(K_{f1}-K_{m}\right)}\right]+\left[\frac{V_{f2}K_{m}\left(K_{f2}-K_{m}\right)}{K_{m}+\beta_{2}\left(1-V_{f2}\right)\left(K_{f2}-K_{m}\right)}\right]\;$$
(5)$$G_{f}=\frac{\;G_{f}}{2\left[1+V_{f}\right]}\;$$
(6)$$G_{m}=\frac{\;G_{m}}{2\left[1+V_{m}\right]}\;$$
(7)$${\bar{G}}_{c}=G_{m}+\left[\frac{V_{f1}G_{m}\left(G_{f1}-G_{m}\right)}{G_{m}+\beta_{1}\left(1-V_{f1}\right)\left(G_{f1}-G_{m}\right)}\right]+\left[\frac{V_{f2}G_{m}\left(G_{f2}-G_{m}\right)}{G_{m}+\beta_{1}\left(1-V_{f2}\right)\left(G_{f2}-G_{m}\right)}\right]\;$$
(8)$$\beta_{1}=\frac{2-5.5{Ѵ}_{m}}{20\left[1-V_{m}\right]}\;$$
(9)$$\beta_{2}=2-\beta_{1}\;$$
where the *V* is the volume fraction and Ѵ is the passion ratio. The subscripts of m and f stand for matrix and fillers. As the prediction is based on dilute distribution which is considered as perfectly bonded and the cut-off limit to use this model will be 30 wt.% as the data gathered from the experimental value shows Young’s modulus started to drop at this point due to the mixture, which is too concentrated. The relation calculated with this new model, shown in Figure 6, shows that the relation between the new model and experimental data best fits the R^2^ is 0.956, which is acceptable, and the unmodified models, which are lower than 0.86.

## 4. Summary

In this work, the mechanical properties, which include Young’s modulus of epoxy-containing both single silica (micro and nano silica) and its hybrid (at 1:1 wt.%), were measured. In the case of Young’s modulus measurement, the addition of a single filler caused a significant drop in the values in the early state, however they started to increase and reached the maximum value at 30 wt.%, regardless of their size. Nano silica particles showed better properties than micro silica for the single filler. In contrast, the addition of a dual filler improves the values of Young’s modulus. At 25 wt.% hybrid (12.5 wt.% micro silica + 12.5 wt.% nano silica), the values of Young’s modulus 228% with less loading compared with single filler which reaches the highest Young’s modulus at 30 wt.%. As demonstrated by other researchers previously, the synergistic effects are also observed in this work due to the different sizes of silica fillers. The new model produces which includes both fillers and epoxy matrix in the equation performs well to predict the value of Young’s modulus which supported by the experimental data.

## Figures and Tables

**Figure 1 polymers-14-03969-f001:**
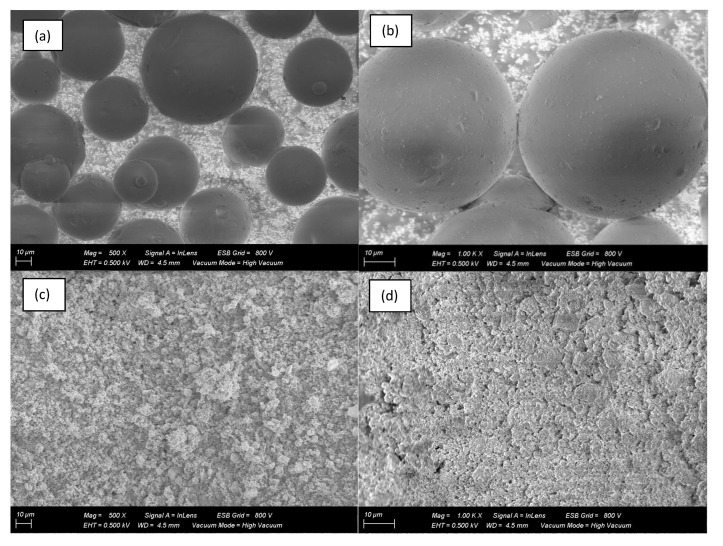
Typical morphology of micro silica particles, with magnification of 500 (**a**), micro silica particles, with magnification of 1000 (**b**), nano silica particles, with magnification of 500 (**c**), and nano silica particles, with magnification of 1000 (**d**).

**Figure 2 polymers-14-03969-f002:**
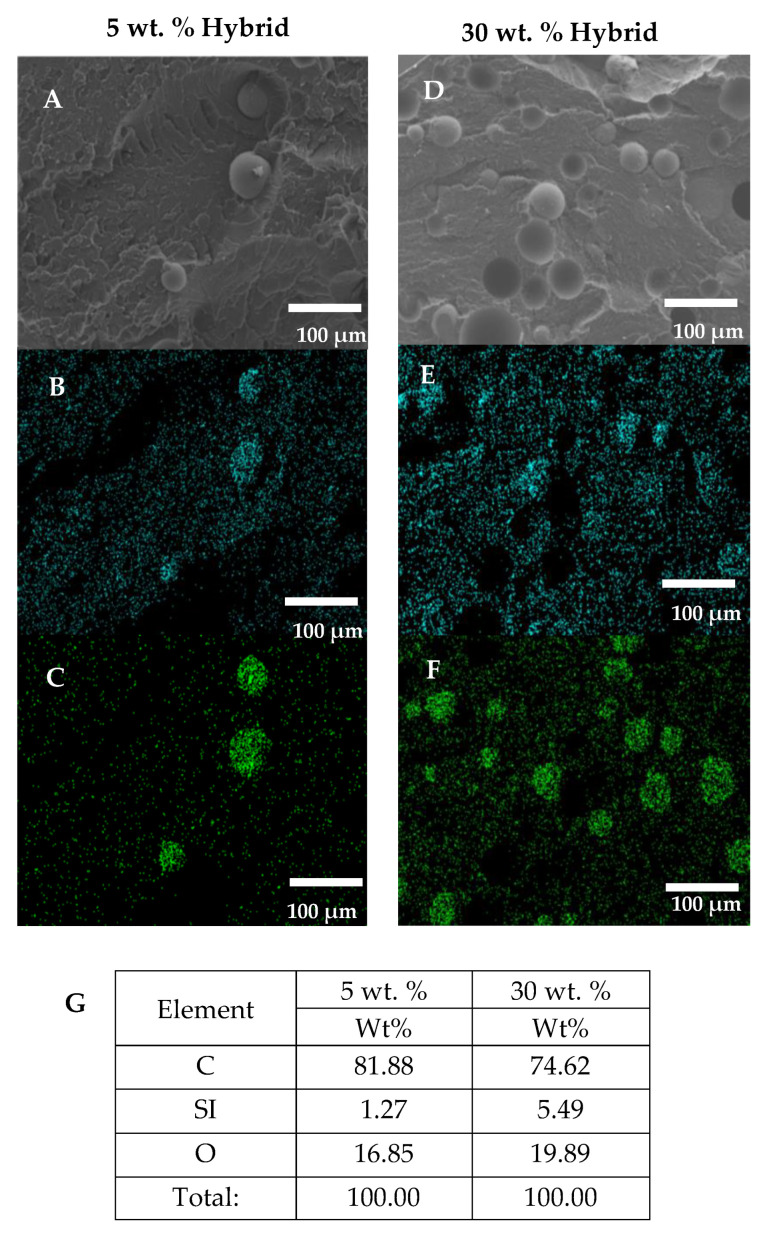
FESEM and EDX analysis of hybrid composites for 5 wt.% (**A**–**C**) and 30 wt.% (**D**–**F**). Figure (**B**,**E**) is showing EDX mapping analysis for silicon for both loadings. Figure (**C**,**F**) is showing EDX mapping analysis for oxygen for both loadings. Weight percent measurement which is obtained from the EDX analysis (**G**).

**Figure 3 polymers-14-03969-f003:**
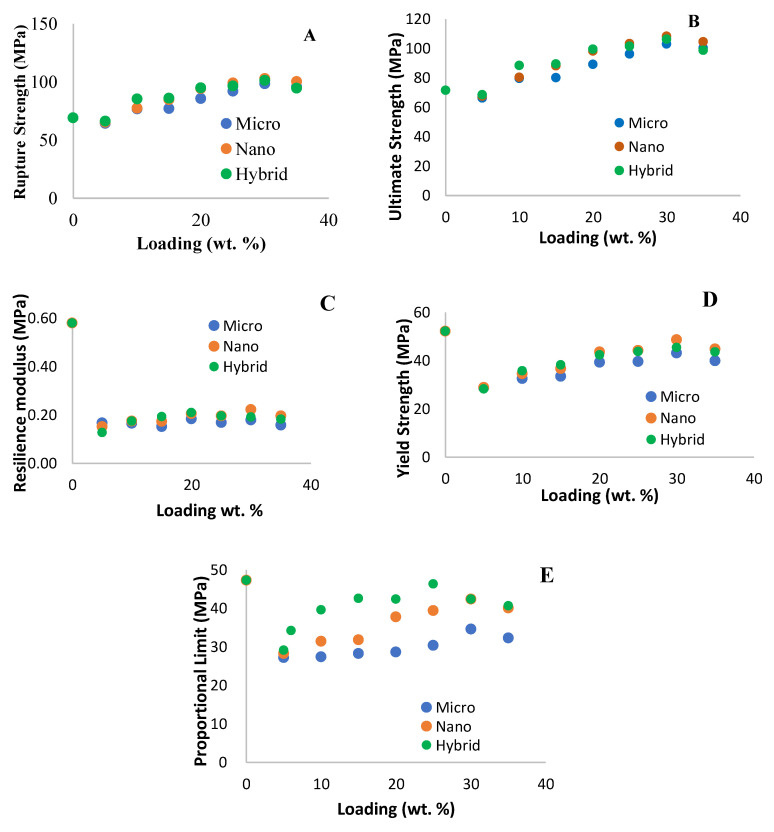
Comparison between Micro, Nano and Hybrid silica composite with respect to Rupture modulus (**A**), Ultimate modulus (**B**), Resilience modulus (**C**), Yield strength (**D**) and Proportional limit (**E**).

**Figure 4 polymers-14-03969-f004:**
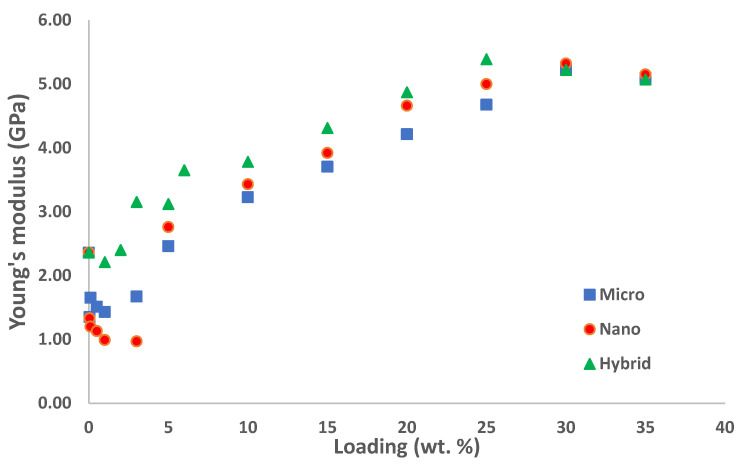
Comparison of Young’s Modulus values between Micro, Nano, and Hybrid Silica.

**Figure 5 polymers-14-03969-f005:**
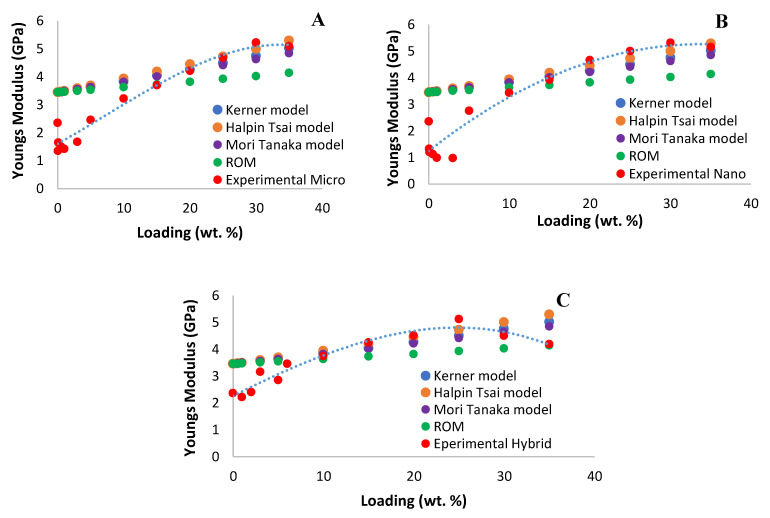
Comparison between the theoretical models and experimental data of epoxy silica filler with different sizes and combinations of fillers as a function of loading wt.%. Where (**A**), is the comparison for Micro silica, (**B**), is the comparison for Nano and (**C**), is the comparison for Hybrid silica.

**Figure 6 polymers-14-03969-f006:**
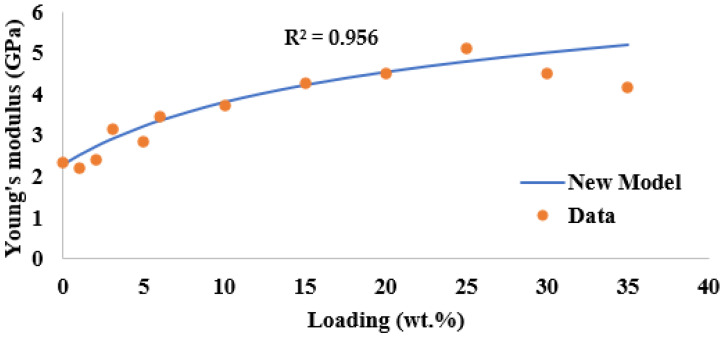
Comparison between the new model and experimental data.

**Table 2 polymers-14-03969-t002:** Young’s modulus test result for micro, nano, and hybrid silica fillers.

Loading (wt.%)	Young’s Modulus, E (GPa)		
	Micro	Nano	Hybrid
0	2.36	2.36	2.36
0.05	1.35	1.33	-
0.1	1.65	1.20	-
0.5	1.51	1.13	-
1	1.43	0.99	2.21
2	-	-	2.40
3	1.67	0.97	3.15
5	2.46	2.76	3.12
6	-	-	3.65
10	3.23	3.43	3.78
15	3.70	3.92	4.31
20	4.21	4.66	4.87
25	4.68	5.00	5.39
30	5.22	5.32	5.22
35	5.10	5.15	5.07

**Table 5 polymers-14-03969-t005:** Comparison of regression (R^2^) between model and experimental data.

Model	Micro	Nano	Hybrid
Kerner	0.92	0.85	0.84
Halpin Tsai	0.93	0.86	0.85
Mori-Tanaka	0.93	0.86	0.85
ROM	0.93	0.86	0.85

## Data Availability

Not applicable.

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
