# Peer review of "The Synergistic Effects of Hybrid Micro and Nano Silica in Influencing the Mechanical Properties of Epoxy Composites—A New Model"

_polymers, 2022, doi:10.3390/polym14193969_

Round 1

Reviewer 1 Report

This article is related to the mathematical model in the mechanical properties of epoxy properties, showing some novelty. But in the article, there are too many problems to support the main idea. We suggest this manuscript should be put forward into major revision. The experiment design and fabrication should be restarted. I base my decision on several issues of which I have summarized the most prominent in a number of points below.

1) In the Introduction section, some expressions are not enough appropriate. In Line 72, the sentence “The result showed tensile strength at a content of 11 w.t% of nano silica increased tensile and flexural” seems ambiguous. Similarly, the sentence in Line 81 “which is higher”, obviously is not an elegant expression. The long sentence beginning at Line 89 is difficult to be understood. Especially, the relation behind the preposition “between” cannot be distinguished. The trace of machine translation is usually found in this article.

 2) In the Introduction section, many reports have not been deeply analyzed and show the mechanism of filler strengthen, living only on the values of mechanical properties. The explanation in Line 45 is not persuasive, and lacks evidence.

3) Table 1 has a number of questionable points. Firstly, nanoparticles have a lower mesh number than microparticles! Secondly, the Line PH shows nanoparticles have a size of ~30 nm. Please explain the concept of PH. I don’t know why several data were missing. For surface area, the value of 450-550 m2/g is such a large value, common micro silica cannot reach this number, which should be explained with suitable evidence. The micrographs of filler particles should be offered in this manuscript.

4) I also questioned the condition of composite fabrication; that is, 500 rmp for 5 min cannot satisfy the requirement of composite preparation, especially in the case of incorporating nanofillers. The manufacturer of epoxy is not displayed in the Experimental section, which is not suitable for scientific writing.

5) In Part 1, Result and Discussion, I found only seven loading values, as well as the fixed ratio of hybrid particles, has been related in Figure 1, which offers not enough support for the main idea. The explanation in Line 158 is not sufficient for lack of effective evidence.

6) For this model, only Young’s modulus is described by mathematical expressions. However, the authors have not shown the constants in all the formulas, which are not in agreement with the values in the Mori Tanaka model. The fitting steps and parameters determination should be emphasized in this article. The transformation from the original model to the modified model should be elucidated in detail. The coefficient R2 in Line 310 is 0.86, and in Figure 4 is 0.956. Which one is right?

7) The most important issue in this paper is the synergistic effect. In fact, the authors showed some favorable results, but all the results are obtained in the case of a 1:1 ratio for micro and nanoparticles. I claim this treatment cannot sufficiently support the model suggested by the authors.

Author Response

Hi, We have submitted the response as in the attachment.

Thank You.

Reviewer 2 Report

This manuscript showed enhanced mechanical properties of epoxy composites by combining hybrid mixtures of micro- and nano-sized silica particles. Below are comments for the revision. 

1. Please, describe physical meanings for following issues. 

(1) Why did authors select a hybrid mixture of micro- and nano-sized Si particles of only "1:1" wt% ratio? (Number densities for micro/nano Si particles will be totally different.) Is there any possibility for epoxy composites to be more increased by adding a different hybrid Si set? (e.g., 0.25: 0.75 wt% ratio for micro and nano Si) 

(2) Why did authors focus on only Young's modulus property among many mechanical properties? (In UTM tests, ultimate strength or elongation at break, etc)

(3) Can authors check the distribution of micro and nano Si particles inside the composite? (This reviewer think that two kinds of particles might not  be uniformly distributed inside the matrix.)  

2. Formulation table should be included.

Information for epoxy resin should be more described.  

Line 113-114: the description for 2:1 combination of epoxy and hardner was somewhat ambiguous. 

In Table 1, actual particle sizes for micro and nano Si particles should be noted. (micron-m or nm...)   

Author Response

(The authors gave the same response as above.)

Reviewer 3 Report

This article presents some interesting data on the complex interactions or micro and nano silica compositions on the mechanical performances of Epoxy Resins. The outcomes are well supported by the data which is presented and prepared in a sound scientific manner. This is a complex and well published field however, and there is already a related article published back in 2012 on this exact subject with similar outcomes. The data in the article here does provide some novel insights and interesting outcomes. Before publication the authors should therefore, justify the novelty of their work in relation to that previously published. Also, the grammar needs some attention. Comparison study of some mechanical properties of micro and nano silica EP composites
  • January 2012
  • DOI: 
  • 10.13140/RG.2.2.23605.42725

Author Response

(The authors gave the same response as above.)

Round 2

Reviewer 1 Report

This article is related to the mathematical model in the mechanical properties of epoxy properties, showing some novelty. I have demonstrated the suggestions to revise the manuscript. But the revision has not satisfied my expectations. Therefore, I regret that I cannot recommend this manuscript for publication. Herein, the main reasons were listed as follows:

1) The key point is the ratio of micro and nano silica fillers was fixed. The new model presented in this manuscript has two volume fractions. They are not independent. So, this model inherently shows a severe defect.

2) Although SEM graphs have been exhibited, the micro silica particles show large dimensions, which is not in agreement with the number of specific surface areas. The nano-silica particles have not clearly been displayed in SEM graphs. Using the agglomerated morphology, the authors mentioned, “nanoparticle cannot be determined exactly” (language error), which is not persuasive to us. Other sample preparation methods should be used.

3) Authors mentioned “complicating crosslink between polymer chains” in Line 193, which shows a misconception because we believe nano silica cannot participate in the crosslinking reactions of epoxy resin.

Author Response

Morng, 

The comment is attached as below

Reviewer 2 Report

Authors fully considered reviewer's comments in the revision. 

Author Response

Morng,

Tq

Reviewer 3 Report

The comments have been fully addressed to my satisfaction.

Author Response

Morng,

The comment are as attachment.

Tq
